# Estimation of the Center of Mass of GRACE-Type Gravity Satellites

**Zhiyong Huang [1,2], Shanshan Li [1,\*], Lin Cai [3], Diao Fan [1] and Lingyong Huang [2]**

[1]    Institute of Geospatial Information, Information Engineering University, Zhengzhou 450001, China
[2]    State Key Laboratory of Geo-Information Engineering, Xi'an 710054, China
[3]    School of Physics, Huazhong University of Science and Technology, Wuhan 430074, China
[\*]    Correspondence: zzy_lily@sina.com

**Abstract:** One of the key constraints for the accelerometer of GRACE-type gravity satellites to accurately measure the non-gravitational accelerations acting on the satellite is that the center of mass of the satellite and the proof mass of the accelerometer should maintain a coincidence. In addition, the accuracy requirement is that the center of mass offset (CM-offset) in the three directions is less than 100 microns. Since the center of mass (CoM) of the satellite will change with the consumption of cold-gas fuel in the tanks, it is necessary to regularly carry out the CoM calibration maneuver. Firstly, the observation equations consisting of the accelerometer linear acceleration, angular acceleration, and the CM-offset vector are established in order to estimate the amount of CM-offset. Then, according to the estimated CM-offset, the satellite mass trim mechanisms are used to change the satellite's CoM, so that the satellite's CoM always approaches the proof mass of the accelerometer, with an accuracy of 100 μm per axis. The CM-offset of the satellite of GRACE-FO is estimated by using the accelerometer, star camera, magnetic torquer, magnetometer, and the precision orbit data during the GRACE-C CM-offset calibration period on 1 February 2020. Four kinds of CM-offset results are obtained by four different angular accelerations as follows: the angular acceleration based on the attitude dynamics ("MTQ angular acceleration"), the accelerometer angular acceleration calibrated by MTQ, the accelerometer angular acceleration, and the angular acceleration calculated by the star camera. By comparing the four kinds of CM-offset results that are estimated by the four different methods, all four of the results are shown to have the same level of accuracy. Based on the accelerometer (calibrated) angular acceleration, the difference with the JPL result is 0.5 μm, while the difference between the conventional method and the JPL result is 6.0 μm. All four of the methods can achieve the requirement of 50 μm accuracy and using four CM-offset estimation methods simultaneously can improve the integrity of the calibration results. Subsequently, the CM-offset results of GRACE-C since its launch are estimated here. The calibration algorithm that is proposed in this paper can be used as a reference in the calibration of gravity satellites carrying an accelerometer payload.

**Keywords:** GRACE; GRACE Follow-On; gravity satellites; center of mass; calibration

## 1. Introduction

The gravity recovery and climate experiment (GRACE), which is a joint mission between the National Aeronautics and Space Administration (NASA) and the German Aerospace Center (DLR), was successfully launched on 17 March 2002 [1,2]. The concept of gravity field measurement by low-low satellite-to-satellite tracking (LL-SST) was realized for the first time. GRACE consists of two polar-orbiting satellites at a height of 500 km, carrying a dual-frequency K/Ka-band microwave ranging system in order to measure the distance changes between the satellites, and an electrostatic suspension accelerometer to measure the non-conservative forces that are acting on the satellites. Combined with the satellite orbit data, the time-variable and the static gravity fields can be recovered. The accuracy of the Earth's gravity field model that was obtained from the 39 days' in-orbit data

exceeds that which was obtained from the data accumulated by all of the space geodetic satellites in the last 30 years [3,4]. More importantly, GRACE has opened a new era in the study of time-variable gravity fields [5]. In view of the great success of GRACE, following it being decommissioned at the end of 2017, the GRACE Follow-On was launched on 22 May 2018 in order to succeed the science mission of GRACE.

In this paper, GRACE and GRACE Follow-On are generally referred to as GRACE-type gravity satellites [6], or simply GRACE-type satellites. GRACE Follow-On's two satellites are called "GRACE-C" and "GRACE-D".

A series of on-orbit calibration of the spacecraft payloads is required in order to ensure that each payload meets the basic conditions of scientific observation and obtains scientific observation data [7]. Once the satellite is launched into orbit, it is necessary to calibrate the center of mass (CoM) offsets, the KBR antenna phase center, the star camera installation frame, the accelerometer calibration parameters, the GNSS antenna phase center, and the other payload parameters of the satellites.

In order to meet the observation conditions of the various payloads of the GRACE-FO satellites, the accuracy of the satellite payload installation parameters was strictly constrained in the initial design stage. In general, in an ideal case, the strict constraint relation of "ten centers are collinear" should be satisfied between the satellite's CoM and the center of accelerometer (ACC) proof mass, the KBR antenna phase center, the center of the twin propellant tanks, and the center of drag (CoD) of the two satellites. As illustrated in Figure 1 (Bandikova (2015) [8]), the "SF" denotes the satellite's body reference frame (satellite frame (SF)) and the origin of the SF is at the target location of the CoM; in other words, the center of ACC proof mass. The SF coordinate axes are defined as follows: $X_{SF}$ points from the origin to the target location of the KBR antenna phase center (roll axis); $Y_{SF}$ forms a right-handed triad with $X_{SF}$ and $Z_{SF}$ (pitch axis); $Z_{SF}$ is normal to the $X_{SF}$ axis and to the plane of the main equipment platform (yaw axis) [7].

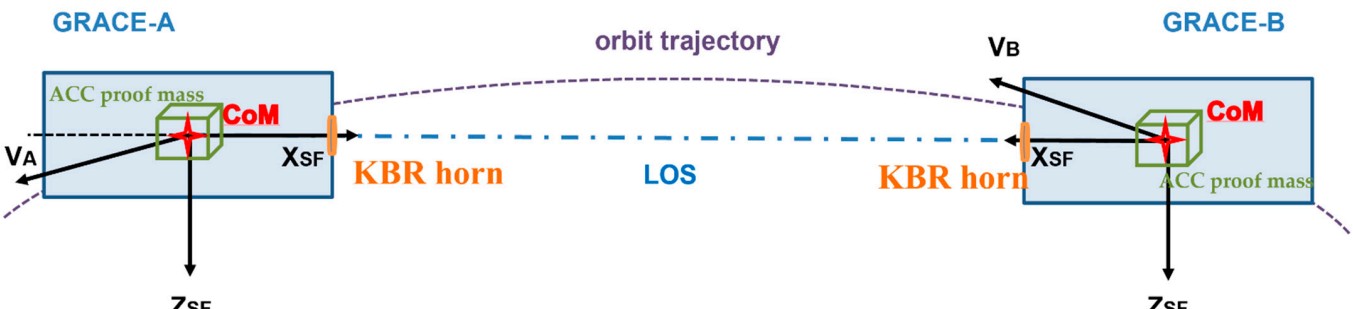

**Figure 1.** Geometric constraints among CoM, ACC proof mass center, and KBR horn of GRACE-A and GRACE-B.

In an ideal situation, the ACC proof mass is precisely located in the satellite's CoM, and the measurement requires that the KBR horn vector origin from the CoM is aligned with the line of sight (LOS, the lines of the satellite's CoM) within a few milliradians. Therefore, the lines between the CoM, the ACC proof mass, and the KBR horn are collinear. This paper focuses on the estimation algorithm in order to determine the CM-offset of the satellites.

In order to meet the normal observation conditions of the ACC, during the initial technical index and the parameter design stage, the offset between the satellite's CoM and the center of ACC proof mass should not exceed 100 μm. Otherwise, due to the gravity gradient and the centrifugal force, the measurements of the ACC angular acceleration will be coupled with the ACC linear acceleration measurements, therefore affecting the linear acceleration observation accuracy. The installation accuracy of the CoM-related parameters was proposed during the satellite installation stage [9], as shown in Table 1. The center of mass offset (CM-offset) [10] is defined as the offset vector from the origin of the center of ACC proof mass to the satellite's CoM in this paper.

**Table 1.** Installation accuracy requirements of CoM-related parameters of GRACE-FO satellites.

| | |
|---|---|
| Center of drag (CoD) | CoD in $\pm$X axis direction (with origin in the spacecraft CoM) with a tolerance of $\pm$500 μm |
| CoM relative to ACC fiducials (ground alignments and alignment knowledge) | 550 μm in all three axes (translational alignment, including knowledge uncertainty) |
| ACC fiducials relative to CoM (in-flight stabilities and center of mass migration) | (1) 9 μm/orbit in all three axes (translational) with frequency content of the amplitude spectrum at 2 cpr less than 2 μm(2) 100 μm/6 months in all three axes (CoM migration) |

Therefore, in order to accurately control the CoM position of the satellite, the mass layout of the GRACE-type gravity satellites is highly balanced, the two propellant tanks are placed symmetrically in the direction of the satellite's flight, and its symmetry point coincides with the satellite's CoM. In addition, in order to minimize the residual atmosphere and the resulting perturbation torque from the thruster firings, the satellite's CoD is designed to be located on the longitudinal axis of the SF and coincident with the geometric center of the KBR horn [9].

The following three factors lead to CM-offset: (1) the initial installation error of CoM on the ground; (2) the CM-offset of the satellite caused by 1 g/0 g effects after the satellite is in orbit; (3) the CM-offset of the satellite in X direction caused by the asymmetrical cold-gas consumption of two propellant tanks. GRACE's CM-offset calibration results show that the CM-offset that is caused by the first two factors is within 300 μm. While in orbit, the CM-offset of the satellite is required to change no more than 100 μm within six months. In order to compensate for the CoM measurement errors on the ground and the CoM changes that are caused by various factors after launch, it is necessary to increase the center of mass trim assembly mechanism (MTM) on all three of the axes of the satellite and to adjust the MTM accordingly.

The estimation of the satellite's CoM mainly uses the in-orbit spin maneuver, which can be divided into two schemes [11–14] according to whether the satellite is equipped with a high-precision ACC. Starting from the geometrical relationship between the thruster installation position and the nominal and actual CoM position, the satellite's CoM was solved by using the attitude dynamics equation [15]. When the satellite is equipped with a high-precision ACC, the CM-offset and the angular acceleration measurements are used to estimate the linear acceleration, and the least-square comparison is made with the linear acceleration measurements that are observed by the ACC in order to achieve the CoM-offset estimation [16–18]. However, the above schemes have high requirements for the thruster installation position and the thrust precision index, therefore, it is difficult to achieve the above spin maneuver using cold-gas propulsion, and the attitude and orbit control system (AOCS) requires high precision.

In view of the above problems, scholars tend to adopt the magnetic torquers (MTQ) maneuver in order to achieve the satellite spin maneuver and have proposed a method that combines a periodic magnetic dipole moment with Earth's magnetic field in order to generate magnetic torque to cause the satellite to generate an attitude maneuver signal. Based on this maneuver signal, Wang et al. (2010) and Wang (2003) [10,19] used attitude dynamics to smooth the satellite attitude in order to obtain the best precision in terms of the satellite attitude, the angular velocity, and the angular acceleration. This method was successfully applied to the estimation of the CM-offset of GRACE satellites. However, this paper does not provide a conclusion as to what kind of angular acceleration information can obtain the best estimation accuracy for CM-offset, which deserves further study.

Li (2009) [20] conducted a mathematical simulation analysis based on Wang (2003) [19] using the batch estimation theory and the Kalman filter algorithm, and the results showed that the CoM calibration accuracy could reach 100 μm during satellite calibration maneuvers in the North Pole. However, the algorithm was not verified based on GRACE satellite data. Wang et al. (2010) [21] and Xin et al. (2013) [22] proposed an algorithm

to estimate the satellite CoM with an ACC and gyroscope, combined with the predictive filtering with Kalman filtering, and estimated the angular acceleration and CM-offset of the satellite in real time by taking the angular acceleration and CM-offset of the satellite as state estimators. This method considers that the moment-of-inertia and mass of the satellite are not accurate and uses predictive filtering in order to estimate the angular acceleration; however, this increases the complexity and does not conform to the reality that the GRACE-type gravity satellites have a high-precision moment-of-inertia matrix and mass parameters. Since GRACE's CM-offset does not need to be known in real time, batch processing is advantageous.

Based on the method that was adopted by Wang et al. (2010), Wang (2003) [10,19], and Li (2009) [20], the calibration method of ACC angular acceleration by "MTQ angular acceleration" is proposed in this paper. Based on the proposed method, the CM-offset was calculated by using the angular acceleration that was calculated by MTQ, the angular acceleration ACC measurements (calibrated by MTQ), the angular acceleration ACC measurements (not calibrated), and the angular acceleration that was calculated by the star camera quaternions. The estimation results were compared in order to obtain the best CM-offset estimation accuracy. The angular acceleration outliers were identified by the comparison of the angular acceleration, which was calculated with the four methods, and coarse error was avoided in the estimation of the CM-offset.

## 2. CM-Offset Calibration Process of Gravity Satellites

### 2.1. CoM Calibration Maneuver Scheme

The CoM calibration (CM-Cal) maneuver scheme of GRACE-type satellites and the periodic maneuver signals [10] are shown in Table 2.

**Table 2.** GRACE-type satellites CM-Cal maneuver design.

| | |
|---|---|
| CM-offset accuracy | Less than 0.1mm in all three directions |
| CM-Cal maneuver oscillation period | 12 s |
| CM-Cal maneuver directions | Separately applied in the roll (X), pitch (Y), and yaw (Z) direction |
| Duration of maneuver in one direction | 180 s |
| The number of maneuvers | 2 roll, 3 pitch, and 2 yaw |
| Total time spent performing the CM-Cal maneuver | About 8 to 12 h |
| Performs payloads | MTQ |
| Waveform | Square wave |
| Adjusting ability of MTMs | $\pm 2$ mm |

When executing the CM-Cal maneuvers, only 180 s of current was applied to the MTQ rods. As shown in Figure 2, the electric currents in torque 2A and 2B had an oscillation period of 12 s. In order to execute a roll maneuver at a high latitude, we applied an electric current in MTQ-Y (torque 2A and 2B) axis, and the geomagnetic field was mainly in the Z direction at high latitudes; therefore, we could obtain a moment of rotation in the roll (X) axis.

The three axes of the MTQ rods were installed in parallel with the three axes of the satellite. The three axes of the magnetic torque rods were all composed of a cylindrical core and two coils. By applying an electric current in a given axial direction, a magnetic dipole moment $m$ in that direction can be created, and when this interacts with the Earth's magnetic field $B$, the desired magnetic torque can be generated $T = m \times B$. The uniaxial magnetic dipole moment of the GRACE satellites ranged from $-30$ to $+30$ Am$^2$, and its axial alignment with the satellite system was 0.06 degrees. The GRACE-FO satellites produced uniaxial magnetic dipole moments ranging from $-27.5$ to $+27.5$ Am$^2$.

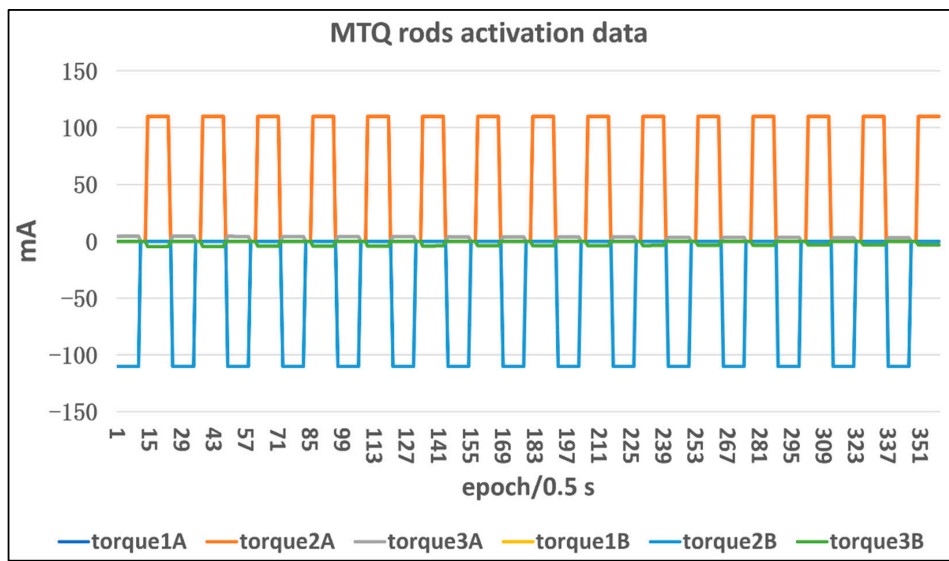

**Figure 2.** MTQ rod activation to achieve the pitch maneuver.

In order to obtain the desired magnetic dipole moment *m*, a specific current can be applied to the three axes of the satellite MTQ rods. For the GRACE/GRACE-FO satellites, the desired magnetic torque and the desired magnetic dipole moment were calculated by the attitude and orbit control system (AOCS) software. All of the data calculations were performed under the science reference frame (SRF) in this paper, whose coordinate origin is defined as the actual CoM position of the satellite, and the origin of the SRF was maintained by regularly performed CM-Cal maneuvers and CoM trim. The CM-offset is a relative quantity, therefore, the definition of the SRF origin does not affect the estimation of the CM-offset. Its coordinate axis is consistent with SF, as shown in Figure 2. In each CM-Cal maneuver, a larger magnetic torque was applied to one axis in order to obtain a larger linear acceleration signal in the other two axes and to estimate their CM-offset. The maneuver along the roll, pitch, and yaw directions were referred to as "roll maneuver", "pitch maneuver", and "yaw maneuver", respectively. For example, if the magnetic torque $T_e$, which is required by the roll maneuver, is the product of $\begin{bmatrix} 1 & 0 & 0 \end{bmatrix}^T \mathrm{N} \cdot \mathrm{m}$ with a square wave function [10], and $T_e$ is the expected torque, we can estimate the CM-offset of the Y and Z directions based on this maneuver.

### 2.2. Location of CM-Cal Maneuvers

Wang et al. (2010) [10] summarized the three main factors that need to be considered in the selection of the maneuvering area in their paper. The first is the desire to maximize the magnitude of the angular acceleration along each axis, since this maximizes the signal for regression, thus improving the calibration accuracy. A second consideration in choosing the maneuver area is to reduce the occurrence of linear acceleration twangs; satellites entering and exiting the Earth's shadowed area are not selected. Since the ground-monitoring station is located in the Northern Hemisphere (Germany), and the angular acceleration is symmetrical with respect to the geomagnetic equator, the northern latitude region is preferred for the calibration maneuvers. The recommended calibration maneuver regions for the GRACE satellites are shown in Table 3.

**Table 3.** GRACE-type satellite CM-Cal maneuver areas.

| Maneuver Axis | Location of CM-Cal Maneuvers |
|---|---|
| Roll: two high-latitude areas | Latitude:60–80°N, Longitude: at random |
| Pitch: two high-latitude areas | Latitude:60–80°N, Longitude: at random |
| Pitch: one low-latitude area | Latitude:10–20°N, Longitude: 35–55°W |
| Yaw: two low-latitude areas | Latitude:10–20°N, Longitude: 120–140°E |

## 3. CM-Offset Estimation Algorithm

The linear acceleration itself is a function of the ACC proof mass offset (the CM-offset) and other variables. If all of the other variables are known or can be fitted with some functions, the ACC proof mass offset can easily be determined from the observations of linear acceleration. These variables include non-conservative forces acting on the satellites, angular velocity, and angular acceleration.

The angular velocity and the angular acceleration are combined with the CM-offset to produce the angular motion disturbance acceleration. The angular velocity and the angular acceleration can be obtained from the quaternion observations of the star camera by fitting these observations into the attitude dynamics equation or the attitude kinematics equation. Since the non-conservative forces that are subjected upon the satellite change very little during the calibration maneuver period, this is smoother compared to the acceleration that is induced by the angular motion that is caused by the magnetic torquer. As a result, the non-conservative forces can easily be separated from the acceleration that is associated with the angular motion that is caused by MTQ, from which the CM-offset can be determined.

### 3.1. Estimation of CM-Offset Based on Least Squares

The SuperSTAR ACC measures the relative acceleration of the proof mass and the electrode cage [10]; that is, the acceleration that is generated by the excitation signal, which can be written as follows:

$$a_{\text{exc}} = -\ddot{d} - \dot{\omega} \times d - 2\omega \times \dot{d} - \omega \times (\omega \times d) + a_{\text{gg}} + a_{\text{ng}} + a_{\text{thr}} \tag{1}$$

In Equation (1), $\omega \times (\omega \times d)$ is the centripetal acceleration, $\dot{\omega} \times d$ is tangential acceleration, $a_{gg}$ is the gravitational gradient, $a_{ng}$ is the non-conservative force, $a_{thr}$ is the residual linear acceleration that is caused by the imperfect attitude maneuver coupling, $d$ is the CM-offset, $\omega$ and $\dot{\omega}$ are the angular velocity and the angular acceleration, respectively, and $\dot{d}$ and $\ddot{d}$ are the first and second partial derivatives of the CM-offset relative to time. The CM-offset is considered to be unchanged in the calibration period, and the first and second partial derivatives are 0. Therefore, the linear acceleration of the excitation signal can be simplified as follows:

$$a_{\text{exc}} = -\dot{\omega} \times d - \omega \times (\omega \times d) + a_{\text{gg}} + a_{\text{ng}} + a_{\text{thr}} \tag{2}$$

As the gravity gradient has long-term characteristics, this was assumed to be constant during the period of the short-term calibration maneuver. The attitude control thrusts are in the off state during the CM-Cal maneuver stage (180 s) and, for the residual attitude maneuver force, only the long-term period exists (attitude maneuver beyond 180 s); this was also assumed to be constant. The frequency band of the non-conservative force is less than 40 mHz and is close to linear term, within 180 s. Therefore, the effect of the non-conservative force can be removed by linear fitting, and only a 0.083 Hz square wave modulation signal by MTQ is retained. Therefore, the linear acceleration $a_{\text{exc}}$ output by the ACC can be expressed as $a_{\text{out}}$ as follows:

$$a_{\text{out}} = -\dot{\omega} \times d - \omega \times (\omega \times d) + \alpha t + \beta + \eta \tag{3}$$

In Equation (3), $\alpha$ and $\beta$ are the linear and the constant deviation unknown terms of the linear acceleration, which can be removed by a "detrend" handle, and $\eta$ is the measured noise, residual unsimulated noise, and the ignored acceleration term.

After substituting the calculated $\omega$ and $\dot{\omega}$ into Equation (3), the basic mathematical model of CoM calibration can be obtained as follows:

$$l = a_{\text{out}} - \alpha t - \beta = \mathbf{A} \cdot d \tag{4}$$

In Equation (4), the measurement matrix $l$ (O–C) is defined as the observed value $a_{\text{out}}$ (O) of the SuperSTAR ACC minus the fitted calculation value $\alpha t + \beta$ (C), and the design matrix $\mathbf{A}$ is represented as follows:

$$\mathbf{A} = \begin{bmatrix} \omega_y^2 + \omega_z^2 & \dot{\omega}_z - \omega_x\omega_y & -(\omega_x\omega_z + \dot{\omega}_y) \\ -(\omega_x\omega_y + \dot{\omega}_z) & \omega_x^2 + \omega_z^2 & \dot{\omega}_x - \omega_y\omega_z \\ \dot{\omega}_y - \omega_x\omega_z & -(\omega_z\omega_y + \dot{\omega}_x) & \omega_x^2 + \omega_y^2 \end{bmatrix} \tag{5}$$

The least square solution to Equation (5) is represented as follows:

$$d = \left(\mathbf{A}^T \mathbf{P} \mathbf{A}\right)^{-1} \mathbf{A}^T \mathbf{P} l \tag{6}$$

According to the ACC measurement accuracy characteristics of the GRACE satellite, we assume that the direction Y (pitch axis) is an insensitive axis with a measurement accuracy of $1 \times 10^{-9}$ m/s$^2$/$sqrt$(Hz) and direction X (roll axis), and Z (yaw axis) is a sensitive axis with a measurement accuracy of $3 \times 10^{-10}$ m/s$^2$/$sqrt$(Hz). If the standard deviation of the prior unit weight is $\sigma = 3 \times 10^{-10}$ m/s$^2$/$sqrt$(Hz), then $p_1 = 1$, $p_2 = 9/100$, $p_3 = 1$, and the corresponding weight matrix is defined as follows:

$$\mathbf{P} = \begin{bmatrix} p_1 & 0 & 0 \\ 0 & p_2 & 0 \\ 0 & 0 & p_3 \end{bmatrix} \tag{7}$$

then the estimation accuracy of the CM-offset can be expressed as follows:

$$\sigma_d^2 = \left(\mathbf{A}^T \mathbf{P} \mathbf{A}\right)^{-1} \sigma_0^2 \tag{8}$$

where $\sigma_0^2$ is the posterior unit weight variance, and

$$\sigma_0 = \sqrt{\frac{(l - \mathbf{A}d)^T \mathbf{P}(l - \mathbf{A}d)}{n - t}} \tag{9}$$

In Equation (9), $n$ represents the total number of observations, $t$ represents the number of necessary observations, and $n - t$ represents the number of redundant observations.

The next section of this paper introduces three calculation methods for angular acceleration.

*3.2. Angular Velocity Reconstruction Using Quaternions*

$q(t)$ stands for the rotation quaternions from the inertial system to the scientific coordinate system. Assuming that the rotation axis is $a\begin{pmatrix} a_x & a_y & a_z \end{pmatrix}^T$ and the rotation angle is $\phi$, then the quaternions can be expressed as follows:

$$q(t) = \begin{bmatrix} q_0 & q_1 & q_2 & q_3 \end{bmatrix}^T = \begin{bmatrix} a\sin(\phi/2) \\ \cos(\phi/2) \end{bmatrix} \tag{10}$$

The rotation matrix $R_i^{SRF}$ from the inertial system to the scientific coordinate system can be obtained from the quaternions as follows:

$$R_i^{SRF} = \begin{bmatrix} 2(q_0^2 + q_1^2) - 1 & 2(q_1q_2 + q_0q_3) & 2(q_1q_3 - q_0q_2) \\ 2(q_1q_2 - q_0q_3) & 2(q_0^2 + q_2^2) - 1 & 2(q_2q_3 + q_0q_1) \\ 2(q_1q_3 + q_0q_2) & 2(q_2q_3 - q_0q_1) & 2(q_0^2 + q_3^2) - 1 \end{bmatrix} \tag{11}$$

Due to the following relationship between the rotation matrix $R_i^{SRF}$ and the angular velocity $\omega$, we obtain the following:

$$\dot{R}_{SRF}^i = \Theta \cdot R_{SRF}^i, \Theta = \begin{bmatrix} 0 & -\omega_z & \omega_y \\ \omega_z & 0 & -\omega_x \\ -\omega_y & \omega_x & 0 \end{bmatrix} \qquad (12)$$

let

$$R_i^{SRF} = \begin{bmatrix} R_{11} & R_{12} & R_{13} \\ R_{21} & R_{22} & R_{23} \\ R_{31} & R_{32} & R_{33} \end{bmatrix}, \dot{R}_{SRF}^i = \begin{bmatrix} \dot{R}_{11} & \dot{R}_{21} & \dot{R}_{31} \\ \dot{R}_{12} & \dot{R}_{22} & \dot{R}_{32} \\ \dot{R}_{13} & \dot{R}_{23} & \dot{R}_{33} \end{bmatrix} \qquad (13)$$

when we multiply Equation (12) by $R_i^{SRF}$, we obtain the following:

$$\dot{R}_{SRF}^i \cdot R_i^{SRF} = \Theta, \begin{bmatrix} \dot{R}_{11} & \dot{R}_{21} & \dot{R}_{31} \\ \dot{R}_{12} & \dot{R}_{22} & \dot{R}_{32} \\ \dot{R}_{13} & \dot{R}_{23} & \dot{R}_{33} \end{bmatrix} \cdot \begin{bmatrix} R_{11} & R_{12} & R_{13} \\ R_{21} & R_{22} & R_{23} \\ R_{31} & R_{32} & R_{33} \end{bmatrix} = \begin{bmatrix} 0 & -\omega_z & \omega_y \\ \omega_z & 0 & -\omega_x \\ -\omega_y & \omega_x & 0 \end{bmatrix} \qquad (14)$$

The above equation can be rearranged to obtain the angular velocity $\omega$ as follows:

$$\begin{cases} \omega_x = \dot{R}_{12} \cdot R_{13} + \dot{R}_{22} \cdot R_{23} + \dot{R}_{32} \cdot R_{33} \\ \omega_y = \dot{R}_{13} \cdot R_{11} + \dot{R}_{23} \cdot R_{21} + \dot{R}_{33} \cdot R_{31} \\ \omega_z = \dot{R}_{11} \cdot R_{12} + \dot{R}_{21} \cdot R_{22} + \dot{R}_{31} \cdot R_{32} \end{cases} \qquad (15)$$

where $R = (R(t_{i+1}) + R(t_{i-1}))/2, \dot{R} = (R(t_{i+1}) - R(t_{i+1}))/\Delta t, \Delta t = t_{i+1} - t_{i-1}$, $i$ is the serial number of epochs. The angular acceleration can be obtained by the difference in the angular velocity.

### 3.3. Estimation of MTQ Angular Velocity and MTQ Angular Acceleration Based on Batch Estimation Theory

The angular velocity and the angular acceleration can be obtained by the proper processing of the quaternions that are measured by the star camera, but more accurate quaternions, angular velocity, and angular acceleration can be obtained by attitude dynamic smoothing.

The kinematics equation of satellite attitude is as follows:

$$\dot{q}(t) = \frac{1}{2}\Omega(\omega)q(t) \qquad (16)$$

In Equation (16),

$$\Omega(\omega) = \begin{bmatrix} 0 & \omega_z & -\omega_y & \omega_x \\ -\omega_z & 0 & \omega_x & \omega_y \\ \omega_y & -\omega_x & 0 & \omega_z \\ -\omega_x & -\omega_y & -\omega_z & 0 \end{bmatrix} \qquad (17)$$

During the satellite CM-Cal maneuver, a periodic magnetic torquer signal is applied to the satellite. Assuming that the satellite is a rigid body, the attitude dynamics equation of the satellite is as follows:

$$\dot{\omega}(t) = J^{-1}(m \times B - \omega \times (J\omega)) \qquad (18)$$

In Equation (18), $J$ is the moment of inertia of the satellite [23], $m \times B$ is the sum of the magnetic torques that are acting on the satellite, and $B$ is the Earth's magnetic field at the satellite position that is obtained based on the geomagnetic model IGRF13 and is converted into vales in the SRF. The satellite level-1B reduced-dynamic orbit and the satellite attitude

that are solved according to the star camera quaternions of level-1B, are used to solve the Earth's magnetic field $B$.

Let the star camera quaternions $q_{obs}$ be the observed value. Then, the observed equation is as follows:

$$q_{obs}(t) = q(t) \tag{19}$$

The state vector $X(t) = [q(t), \omega(t)]^T$ is defined, and the corresponding state residual is $x(t) = [\Delta q(t), \Delta \omega(t)]^T$. The Taylor series expansion is performed at the estimated state vector $X^*(t) = [q^*(t), \omega^*(t)]^T$; then,

$$q_{obs}(t) - q(t) = \frac{\partial q(t)}{\partial X^*(t)} \cdot x(t) = \left[ \frac{\partial q(t)}{\partial q^*(t)} \quad \frac{\partial q(t)}{\partial \omega^*(t)} \right] \cdot x(t) = \left[ I \quad 0 \right]_{4\times 7} \cdot x(t) = \tilde{\mathbf{H}}_{4\times 7} \cdot x(t) \tag{20}$$

In the above formula, the number of state residuals increases with the increase in the number of observed epochs, resulting in there being no solution to the equation. The state residual can be converted to the initial state vector $x_0 = [\Delta q_0, \Delta \omega_0]^T$; therefore, we obtain the following:

$$x(t) = \mathbf{\Phi}(t, t_0) \cdot x_0 \tag{21}$$

where $\mathbf{\Phi}(t, t_0)$ is the state transition matrix. In this case, the observation equation can be expressed as follows:

$$\Delta q = q_{obs}(t) - q(t) = \tilde{\mathbf{H}}_{4\times 7} \cdot \mathbf{\Phi}(t, t_0)_{7\times 7} \cdot x_0 = \mathbf{H}_{4\times 7} \cdot x_0 \tag{22}$$

The derivative of both sides of Equation (21), with respect to time, can be obtained by the following:

$$\dot{x}(t) = \dot{\mathbf{\Phi}}(t, t_0) \cdot x_0 \tag{23}$$

Then, if $\dot{x}(t) = \mathbf{A}(t) \cdot x(t)$, we obtain,

$$\dot{x}(t) = \mathbf{A}(t) \cdot x(t) = \mathbf{A}(t) \cdot \mathbf{\Phi}(t, t_0) \cdot x_0 \tag{24}$$

By comparing Equations (23) and (24), the state transition matrix is as follows:

$$\dot{\mathbf{\Phi}}(t, t_0) = \mathbf{A}(t) \cdot \mathbf{\Phi}(t, t_0), \mathbf{\Phi}(t_0, t_0) = \mathbf{I} \tag{25}$$

Supposing that $\dot{X}(t) = (\dot{q}(t), \dot{\omega}(t))$, the coefficient matrix of the state transition matrix $\mathbf{A}$ is as follows:

$$\mathbf{A} = \left[ \frac{\partial \dot{X}}{\partial X} \right] = \begin{bmatrix} \frac{\partial \dot{q}}{\partial q} & \frac{\partial \dot{q}}{\partial \omega} \\ \frac{\partial \dot{\omega}}{\partial q} & \frac{\partial \dot{\omega}}{\partial \omega} \end{bmatrix} = \begin{bmatrix} \frac{1}{2}\Omega(\omega)_{4\times 4} & \Gamma_{4\times 3} \\ 0_{3\times 4} & -J^{-1}Q \end{bmatrix} \tag{26}$$

In Equation (26), $Q$ and $\Gamma$ are defined as follows:

$$Q = \Pi(\omega) \cdot J - \Pi(J\omega) \tag{27}$$

$$\Gamma = \frac{1}{2} \begin{bmatrix} q_4 & -q_3 & q_2 \\ q_3 & q_4 & -q_1 \\ -q_2 & q_1 & q_4 \\ -q_1 & -q_2 & -q_3 \end{bmatrix} \tag{28}$$

In the anti-symmetric matrix operator $\Pi()$, the operation on any vector $v = [v_x, v_y, v_z]^T$ is defined as follows:

$$\Pi(v) = \begin{bmatrix} 0 & -v_z & v_y \\ v_z & 0 & -v_x \\ -v_y & v_x & 0 \end{bmatrix} \tag{29}$$

In order to solve the state vector $X(t) = [q(t), \omega(t)]^T$ and state transition matrix $\boldsymbol{\Phi}(t, t_0)$, an 8-order Runge–Kutta integrator can be used according to the variational Equations (16), (18) and (25), respectively. After the state vector and the state transition matrix are known, the observation Equation (22) can be solved with batch estimation theory, and its solution is as follows:

$$x_0 = \begin{bmatrix} \Delta q_0 \\ \Delta \omega_0 \end{bmatrix} = \mathbf{H^T P H} \cdot \mathbf{H^T P} \Delta q \tag{30}$$

where $\mathbf{P}$ is the weight matrix of the observed value, which is an identity matrix in this paper.

After the initial state vector correction $x_0 = [\Delta q_0, \Delta \omega_0]^T$ is obtained, the state vector of the first epoch is adjusted by $X(t_0) = [q(t_0), \omega(t_0)]^T + x_0$, and the numerical integration is performed on Equation (18) again, the angular velocity that is smoothed by the attitude dynamics can be obtained, and the angular acceleration of the attitude dynamics is obtained by directly calculating Equation (18). Since the angular velocity and the angular acceleration of the satellite are mainly driven by the torque, which is generated by the combined action of the magnetic torquer and the geomagnetic field, they are called "MTQ angular velocity" and "MTQ angular acceleration" in this paper.

### 3.4. Calibration of ACC Angular Acceleration

After the high-precision ACC is placed into orbit with the satellite, a series of calibrations is required in order to obtain the correct ACC bias and scale factor. However, in general, the bias and the scale factor of the linear acceleration are more concerning, because the linear acceleration is the measurement of the non-conservative force, which directly affects the recovery accuracy of Earth's gravitational field. In the estimation of the CM-offset, the magnitude of the angular acceleration directly affects the result. Therefore, the MTQ angular acceleration can be used to calibrate the measurements of the ACC angular acceleration in order to obtain the bias deviation and the scale factor.

The MTQ angular acceleration is expressed as $\dot{\omega}_{MTQ}$, the ACC measured angular acceleration is expressed as $\dot{\omega}_{ACC}$, and the angular acceleration scale and the bias factor of the ACC are expressed as $S$ and $B$; therefore, the following calibration equation can be established [24]:

$$\dot{\omega}_{MTQ} = S\dot{\omega}_{ACC} + B + \sum_{n=1}^{N} [A_n \sin(n\omega t) + B_n \cos(n\omega t)] + \dot{\omega}_{noise} \tag{31}$$

where $A_n$ and $B_n$ are the coefficients of the Fourier correction term, and $\omega = 2\pi/T$ is the angular velocity of the satellite orbit motion, which is 97.13 min according to the satellite ephemeris data, and $N$ is the Fourier expansion series. In this paper, $N$ is 2 and $\dot{\omega}_{noise}$ represents the MTQ angular acceleration noise.

The standard least squares were used for the angular acceleration calibration in order to minimize the objective function $\boldsymbol{\Omega} = \|\mathbf{A}x - \mathbf{L}\|^2$, where

$$\mathbf{A} = \begin{bmatrix} \dot{\omega}_{ACC} & 1 & \sin(\omega t) & \sin(n\omega t) & \cos(\omega t) & \cos(n\omega t) \end{bmatrix} \tag{32}$$

The MTQ angular acceleration measurement is expressed as $\mathbf{L}$. The standard least squares solution is $x = \left(\mathbf{A}^T \mathbf{A}\right)^{-1} \mathbf{A}^T \mathbf{L}$, which contains the scale factor $S$ and the bias factor $B$, as well as the Fourier correction coefficients of $A_n$ and $B_n$.

The ACC angular acceleration after calibration is expressed as follows:

$$\dot{\omega}_{cal} = S\dot{\omega}_{ACC} + B + \sum_{n=1}^{N} [A_n \sin(n\omega t) + B_n \cos(n\omega t)] \tag{33}$$

The MTQ angular acceleration can be obtained using the attitude dynamics Equation (18); the ACC measurements contain angular accelerations; the calibrated ACC angular acceleration can be obtained using Equation (33); the angular acceleration that is based

on the star camera quaternions can be obtained by Equation (15); therefore, four different types of angular acceleration can be obtained.

## 4. Estimation CM-Offset of GRACE-FO

According to the sequence-of-events (SOE) logs in the newsletter published by the Jet Propulsion Laboratory (JPL), the official agency of GRACE-FO, GRACE-C, and GRACE-D have each conducted about twenty CoM calibration events. According to the TN-01a_SCE file, which was released by JPL, GRACE-C has four CoM trim (CM-trim) events from 23 May 2018 to 20 February 2022, as shown in Table 4, while GRACE-D did not have any CM-trim events.

**Table 4.** CM-trim events of GRACE-C.

| Date | CM-Trim of GRACE-C |
|------|------------------|
| 18 July 2018 | Y: +103.0 μm |
| 24 July 2018 | Z: −30.0 μm |
| 6 February 2020 | X: −113.0 μm |
| 14 May 2020 | Z: −32.0 μm |

GRACE-FO performed the satellite's first CM-Cal maneuver on 6 June 2018. On 1 February 2020, the CM-Cal maneuver of GRACE-C was carried out, and on 6 February 2020, the CoM of GRACE-C in the X direction was adjusted to −113 μm. This paper focuses on an analysis of the 1 February 2020 calibration data. Firstly, the magnetometer measurements on 1 February 2020 and the IGRF13 geomagnetic model were compared, and then the attitude dynamics of the star camera quaternions fitting residual, the angular velocity, and the angular accelerations were obtained. Then, the 1 February 2020 CM-Cal maneuver data were used to calculate the CM-offset based on four kinds of angular acceleration. Finally, the CM-offset estimation results of GRACE-C since 6 June 2018 were obtained.

GRACE-FO level-1B GNV1B, SCA1B, MAG1B, CLK1B, TIM1B data, and level-1A ACC1A data, including CM-Cal maneuver events from 6 June 2018 to 20 February 2022, were downloaded from the GFZ website (ftp://isdcftp.gfz-potsdam.de (accessed on 15 March 2022)). Their meanings are shown in Table 5.

**Table 5.** GRACE-FO data product for CM-offset estimation.

| Data | Description |
|------|-------------|
| GNV1B | Level-1B reduced-dynamic orbit data, 1 Hz |
| SCA1B | Level-1B compressed/combined star camera data, 1 Hz |
| MAG1B | Level-1B magnetic torque rod activation data + magnetometer data, 2 Hz |
| CLK1B | Clock offset values to convert time tags to GPS Time, 0.1 Hz |
| TIM1B | Mappings from onboard computer to receiver time, 0.125 Hz |
| ACC1A | Level-1A linear acceleration and angular acceleration measurements of the ACC proof mass in AF (ACC Frame), 10 Hz |

Based on the above discussion, a flowchart of the CM-offset estimation, based on four kinds of angular acceleration, was drawn, as shown in Figure 3.

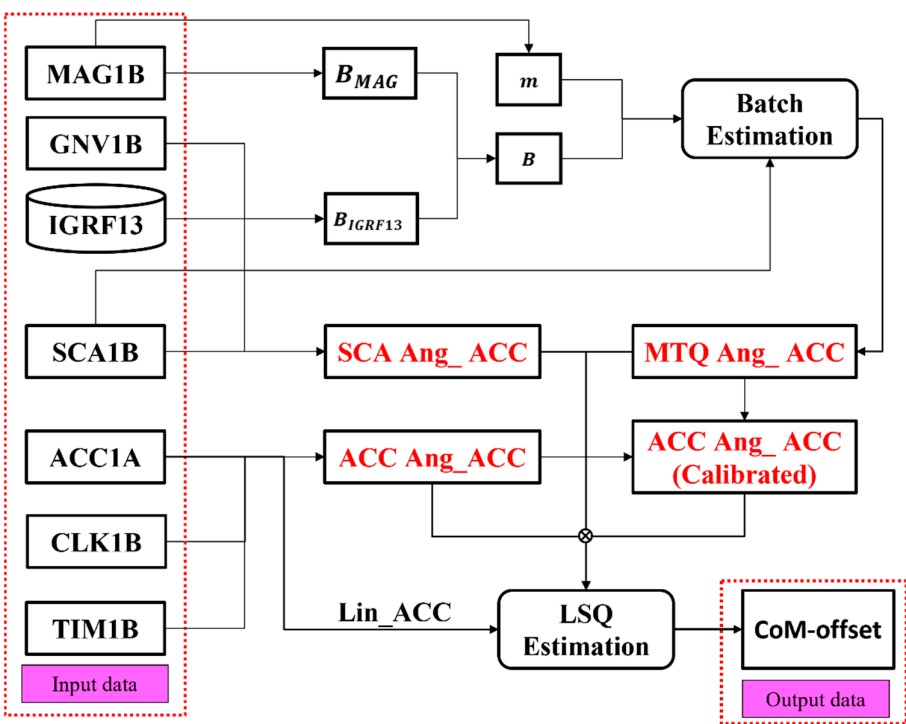

**Figure 3.** Flowchart of the CM-offset estimation based on four kinds of angular acceleration.

## 4.1. Comparison between IGRF13 and Magnetometer Measurements

As part of the AOCS, GRACE-FO carries two fluxgate magnetometers (FGM), one orbiting FGM-A and another as a redundant backup FGM-B [25]. The MAG1B file released by JPL contains the magnetometer data in SRF (unit: mT) and two sets of magnetic torquers activate the current data in SRF (unit: mA) [25]. The IGRF13 geomagnetic model was used to calculate the geomagnetic field data of 1 February 2020, in the local north coordinate system, and then to convert these to the Earth-Fixed Frame, the Inertial Frame, and the SRF, and to compare them with the magnetometer data in the MAG1B file. The difference between the geomagnetic field that is calculated by IGRF13 and the measurements in MAG1B is shown in Figure 4.

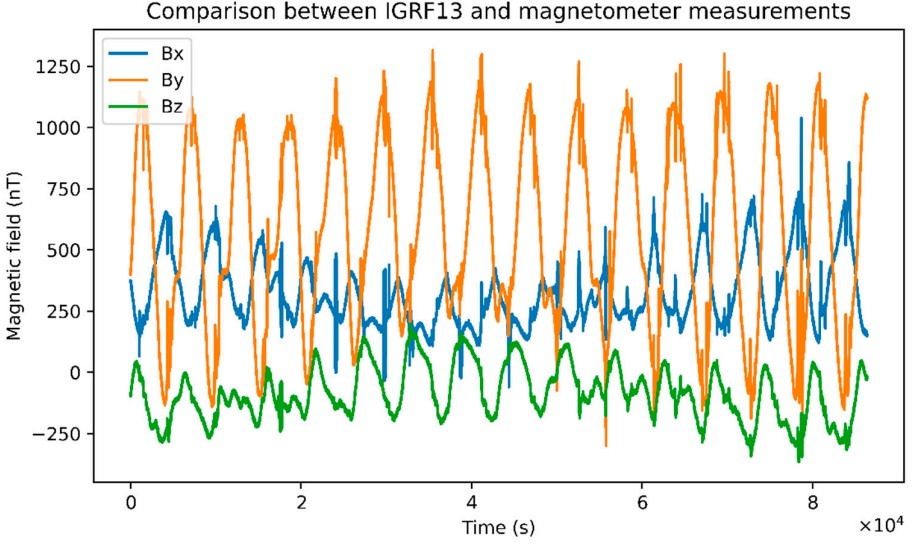

**Figure 4.** Comparison of GRACE-FO magnetometer measurements on 1 February 2020 with IGRF13.

It can be seen from Figure 4 that the maximum difference between the calculated value of the geomagnetic model and the measurements is no more than 1400 nT. At the moment of the CM-Cal maneuver, the magnetometer usually stops working or is switched on intermittently. At this time, the calculated values of the geomagnetic model from IGRF13 can be used to replace the magnetometer measurements for subsequent calculations.

### 4.2. Residual Error of Quaternions Fitting

By selecting the payload data of the first CoM-Cal maneuver period on 1 February 2020 (04:52:49–04:55:49), taking the star camera quaternions as the observed value (O) and the star camera quaternions that were obtained by the attitude dynamics integral as the calculated value (C), the star camera quaternions residual sequence (O–C) can be constructed. The batch estimation theory attributes the O–C residuals of the observation value to the inaccuracy of the initial state vector or other parameters and calculates the parameter correction according to the least squares principle.

The estimated parameters in this paper are only the initial state vectors (the first star camera quaternions, which are the starting point of a numerical integrator). Before (priori residuals error) and after (posteriori residuals error) the initial values of quaternions were corrected, the O–C residual sequence of the star camera quaternions was obtained, as shown in Figure 5. The RMS of the O–C residual sequence of the star camera quaternions before and after the correction of the initial value were all in the order of $10^{-6}$. The priori residual errors of q0–q3 were $2.08 \times 10^{-6}$, $1.72 \times 10^{-6}$, $1.14 \times 10^{-6}$, and $1.81 \times 10^{-6}$, indicating that the initial state parameters and the attitude dynamic parameters are quite accurate, therefore, the attitude dynamic parameters were not estimated any further in this paper. The posteriori residual errors of q0–q3 were $1.96 \times 10^{-6}$, $1.65 \times 10^{-6}$, $1.10 \times 10^{-6}$, and $1.73 \times 10^{-6}$, indicating that updating the initial state parameters can improve the fitting accuracy, and the attitude dynamics fitting method that was adopted in this paper is of high accuracy.

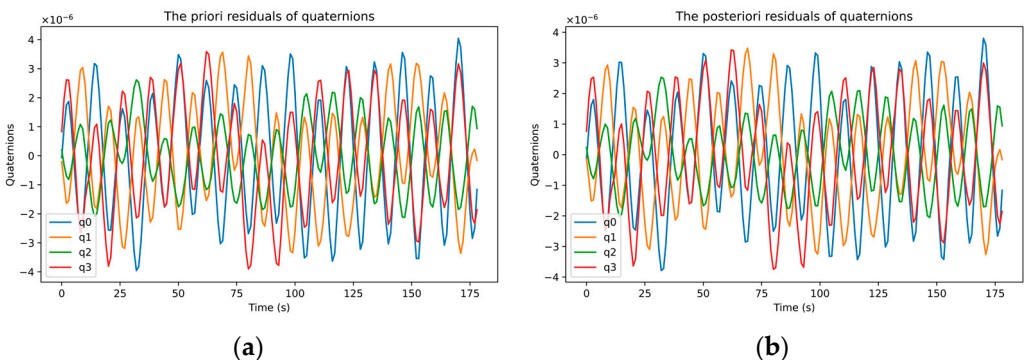

(**a**)  (**b**)

**Figure 5.** Residual errors sequence of attitude dynamics' fitting of star camera quaternions: (**a**) the priori residuals of quaternions; (**b**) the posteriori residuals of quaternions.

### 4.3. Comparison of Four Kinds of Angular Acceleration

According to the attitude dynamics integration and the batch estimation algorithm, the angular velocity and the angular acceleration signals that were applied by the magnetic torquer (MTQ) at GRACE-C satellite's first CM-Cal maneuver time on 1 February 2020 were calculated, as shown in Figure 6.

By removing the constant deviation and the trend items and interpolating them to the ACC linear acceleration acquisition time (ACC1A + TIM1B + CLK1B, 10 Hz), a FIR low-pass filtering was performed at 0.166 Hz (the maneuvering frequency of the CM-Cal maneuver was 1/12 Hz) in order to obtain the attitude dynamic angular velocity and the angular acceleration ("MTQ angular velocity" and "MTQ angular acceleration" in this paper), as shown in Figure 7.

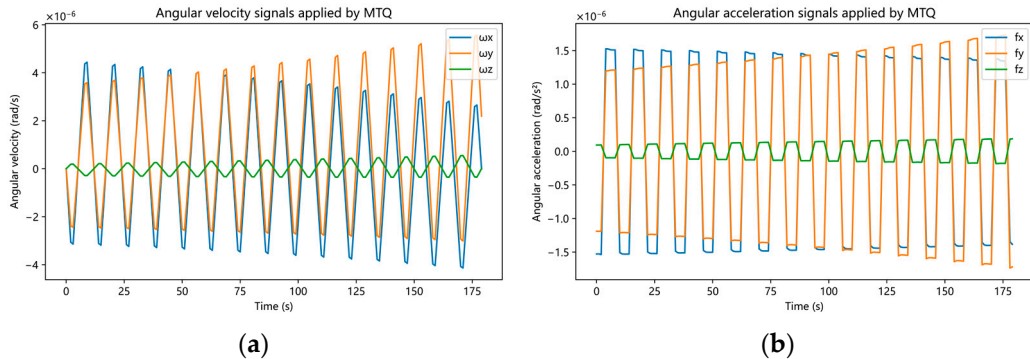

**Figure 6.** Angular velocity and angular acceleration signals applied by MTQ during the first CM-Cal maneuver: (**a**) angular velocity; (**b**) angular acceleration.

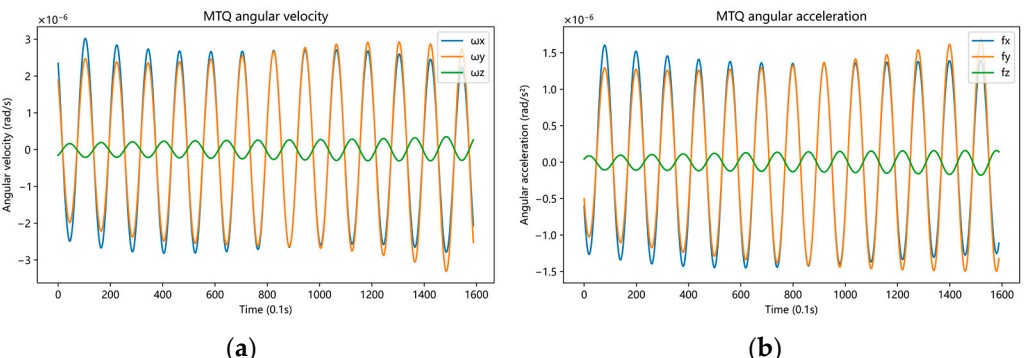

**Figure 7.** Angular velocity and angular acceleration of MTQ are interpolated after filtering: (**a**) MTQ angular velocity; (**b**) MTQ angular acceleration.

At the same time, the star camera angular acceleration (Figure 8a, "SCA angular acceleration" in this paper) was obtained according to Equation (15) using SCA1B data. The angular acceleration of ACC (Figure 8b, referred to as "ACC Angular acceleration" in this paper) was obtained from ACC1A data.

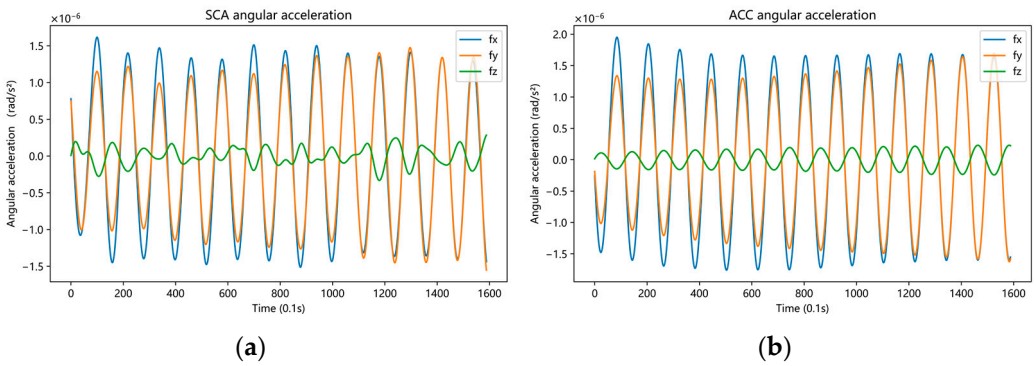

**Figure 8.** Interpolated and filtered SCA angular acceleration and ACC angular acceleration: (**a**) SCA angular acceleration; (**b**) ACC angular acceleration.

The first CoM-Cal maneuver was located at a low latitude, and there were north and radial magnetic fields, which simultaneously generated angular acceleration signals in the roll and pitch directions. As can be seen from the SCA angular acceleration, there is a certain noise in the yaw direction, the ACC angular velocity and the MTQ angular acceleration are relatively smoother, and the angular acceleration in the roll and pitch direction, as measured by ACC, is the largest.

According to the calibration rules, which were proposed in Section 3.4, the ACC angular acceleration measurements were calibrated by the MTQ angular acceleration. As

shown in Figure 9, the magnitude of the calibrated ACC angular acceleration measurements was consistent with the MTQ angular acceleration in three directions. The application of this algorithm was based on the premise that the angular acceleration measured by ACC is not calibrated.

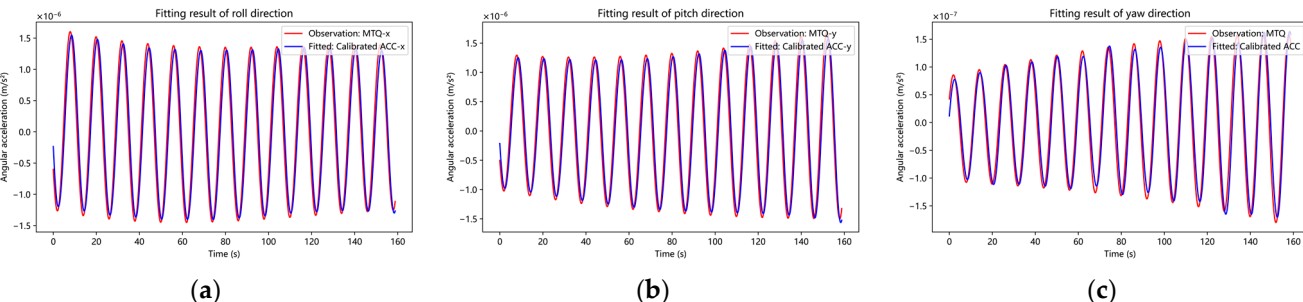

**Figure 9.** Calibration of angular acceleration measurements of ACC: (**a**) roll; (**b**) pitch; (**c**) yaw.

The four angular acceleration signals are inconsistent in phase, although time systems were unified to GPS time according to TIM1B and CLK1B, which may be caused by the inconsistent response delay of the different payloads to the maneuvering signals. Since the ACC angular acceleration and the linear acceleration were calculated from the electrode voltage data, there were no time-scale inconsistencies between them. Although the MTQ angular acceleration is the maneuver signal of the CM-Cal maneuver, there was a time-scale difference between the MTQ angular acceleration and the ACC linear acceleration. If the linear acceleration is regarded as the observed value (O), and the excitation signal is generated by the angular velocity, the angular acceleration and the CM-offset is regarded as the calculated value (C), then, theoretically, the use of the ACC angular acceleration could improve the O–C time-scale consistency. However, there may be inconsistencies between the MTQ angular acceleration and the ACC angular acceleration when the ACC is not calibrated or if some axial acceleration is abnormal after the satellite enters orbit. In this case, the four angular accelerations would be compared in order to obtain the optimal CM-offset estimation result.

### 4.4. CM-Offset Estimation Based on Four Kinds of Angular Acceleration

Based on MTQ, ACC (calibrated), ACC, and SCA angular acceleration, the estimation results of the CM-offset on 1 February 2020 are shown in Table 6. The calculated results of the angular acceleration based on ACC (calibrated), MTQ, and ACC are relatively close to the official calibration result of 113 μm. In terms of formal error, the calibration results based on the ACC angular acceleration had the highest accuracy, while the calibration result based on the SCA angular acceleration had the lowest accuracy. Based on the ACC (calibrated) angular acceleration, the difference compared to the JPL result is 0.5 μm; however, the difference between the conventional method (ACC or MTQ angular acceleration) and the JPL result was 6.0 μm.

**Table 6.** Estimation of GRACE-C CM-offset based on four different angular accelerations.

| CM-Offset | dx (μm) | dy (μm) | dz (μm) |
|---|---|---|---|
| MTQ angular acceleration | $107.2 \pm 0.7$ | $3.7 \pm 0.3$ | $12.4 \pm 0.08$ |
| ACC angular acceleration (calibrated) | $113.5 \pm 0.5$ | $4.2 \pm 0.2$ | $13.2 \pm 0.06$ |
| ACC angular acceleration | $107.0 \pm 0.4$ | $3.6 \pm 0.2$ | $11.5 \pm 0.05$ |
| SCA angular acceleration | $91.6 \pm 1.7$ | $2.6 \pm 0.6$ | $10.4 \pm 0.20$ |

In order to study the calibration results based on seven CM-Cal maneuvers, the results that were obtained by the MTQ, ACC (calibrated), ACC, and SCA angular acceleration are shown in Table 7. It can be seen that the calibration results that were calculated in the yaw

direction and the low-latitude pitch direction deviated from those in the other directions, indicating that the accuracy of the low-latitude calibration is low. Therefore, this paper mainly trusts the calibration results of the two high-latitude pitch and roll directions and takes these as the final results of Table 6 after calculating the weighted average of the results in Table 7.

**Table 7.** CM-offset based on MTQ, ACC (calibrated), ACC, and SCA angular acceleration.

| Maneuver Direction | Angular Acceleration | dx (µm) | dy (µm) | dz (µm) |
|---|---|---|---|---|
| Pitch (low-latitude) | MTQ | 122.7 ± 7.5 | 79.3 ± 24.7 | 9.7 ± 1.0 |
| | ACC (calibrated) | 117.6 ± 7.2 | 42.3 ± 23.6 | 12.6 ± 0.9 |
| | ACC | 112.2 ± 6.8 | 39.4 ± 18.8 | 7.8 ± 0.9 |
| | SCA | 50.1 ± 7.8 | −102.4 ± 23.1 | 10.6 ± 1.2 |
| Pitch | MTQ | 109.5 ± 1.8 | | 22.9 ± 1.8 |
| | ACC (calibrated) | 116.2 ± 1.8 | - | 21.0 ± 1.8 |
| | ACC | 105.6 ± 1.8 | | 23.1 ± 1.7 |
| | SCA | 91.9 ± 2.4 | | 16.6 ± 2.4 |
| Pitch | MTQ | 106.9 ± 0.6 | | 20.3 ± 0.6 |
| | ACC (calibrated) | 113.4 ± 0.4 | - | 21.0 ± 0.4 |
| | ACC | 107.0 ± 0.4 | | 19.7 ± 0.4 |
| | SCA | 91.5 ± 1.5 | | 18.6 ± 1.5 |
| Roll | MTQ | | 4.4 ± 0.3 | 12.7 ± 0.1 |
| | ACC (calibrated) | - | 5.0 ± 0.2 | 13.5 ± 0.1 |
| | ACC | | 4.2 ± 0.2 | 11.5 ± 0.1 |
| | SCA | | 3.2 ± 0.6 | 10.6 ± 0.2 |
| Roll | MTQ | | 3.0 ± 0.3 | 12.0 ± 0.1 |
| | ACC (calibrated) | - | 3.5 ± 0.2 | 12.8 ± 0.1 |
| | ACC | | 3.1 ± 0.2 | 11.4 ± 0.1 |
| | SCA | | 2.0 ± 0.6 | 10.3 ± 0.2 |
| Yaw (low-latitude) | MTQ | −6.0 ± 1.2 | 1.8 ± 0.7 | 8.8 ± 0.6 |
| | ACC (calibrated) | −12.2 ± 1.0 | 1.7 ± 0.6 | 9.0 ± 0.5 |
| | ACC | −10.5 ± 0.9 | 1.3 ± 0.5 | 7.8 ± 0.5 |
| | SCA | −11.0 ± 1.0 | 1.7 ± 0.6 | 8.8 ± 0.5 |
| Yaw (low-latitude) | MTQ | −16.6 ± 1.6 | 3.7 ± 0.7 | 4.4 ± 0.7 |
| | ACC (calibrated) | −20.4 ± 1.2 | 4.1 ± 0.5 | 6.7 ± 0.5 |
| | ACC | −18.8 ± 1.1 | 3.3 ± 0.4 | 5.9 ± 0.4 |
| | SCA | −18.9 ± 1.2 | 3.9 ± 0.5 | 6.1 ± 0.5 |

The MTQ, ACC (calibrated), ACC, and SCA angular accelerations of the second (pitch) and fourth (roll) CM-Cal maneuvers are shown in Figure 10.

In the second (pitch) and fourth (roll) CM-Cal maneuvers, the fitting curves of linear acceleration, when the MTQ angular velocity and the ACC (calibrated) angular acceleration was adopted, are shown in Figure 11. "Observation" represents the linear acceleration measurements of ACC; "Filtered" represents the linear acceleration measurements that were low-pass-filtered with a cut frequency of 0.166 Hz.; "Fitted" represents **A** *d* in Equation (4). The fitting curves that were obtained by the other three methods are similar, therefore, they are not listed. The square root PSD of the fitting curves are shown in Figure 11, and strongest maneuvering signal was found at the frequency of 0.083 Hz. Figure 11d shows that the Z-direction fits well, because the CM-offset in the X-direction is large. However, the other directions (Figure 11c,g,h) failed to reach the maximum amplitude at the maneuvering frequency due to their small CM-offset.

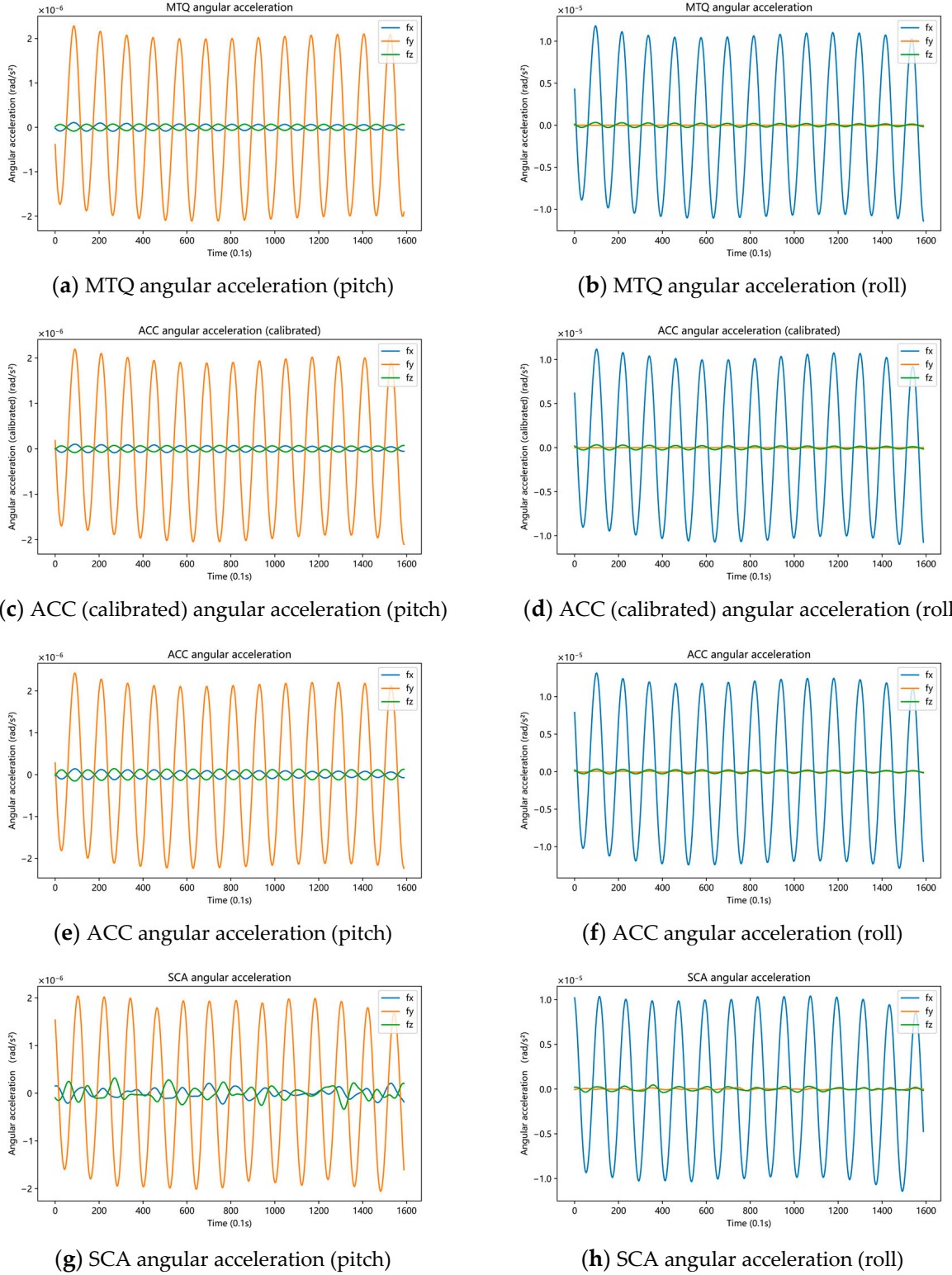

**Figure 10.** Four kinds of angular acceleration.

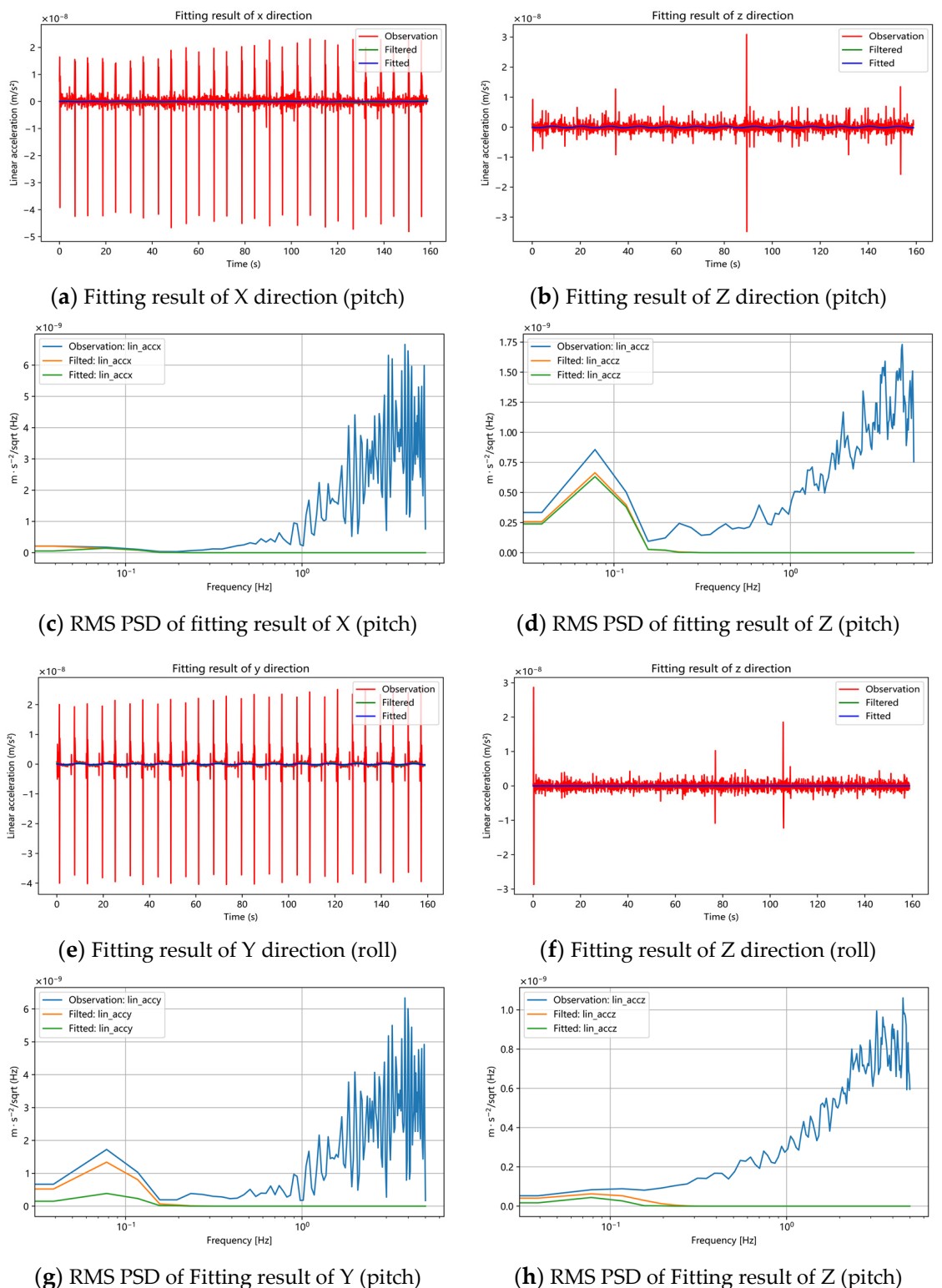

**(a)** Fitting result of X direction (pitch)

**(b)** Fitting result of Z direction (pitch)

**(c)** RMS PSD of fitting result of X (pitch)

**(d)** RMS PSD of fitting result of Z (pitch)

**(e)** Fitting result of Y direction (roll)

**(f)** Fitting result of Z direction (roll)

**(g)** RMS PSD of Fitting result of Y (pitch)

**(h)** RMS PSD of Fitting result of Z (pitch)

**Figure 11.** Fitting cures of linear acceleration in X (**a,c**) and Z (**b,d**) directions during the second (pitch) calibration maneuver and in Y (**e,g**) and Z (**f,h**) directions during the fourth (roll) calibration maneuver.

### 4.5. Estimation of CM-Offset of GRACE-FO

According to the ACC angular acceleration and the MTQ angular velocity that were obtained by attitude dynamics smoothing, the estimation results of the CM-offset on 6 June 2018 are shown in Table 8. Since the first CoM calibration result was not released

by the official agency, the CM-offset estimation results based on this paper show that the CoM changes that were caused by the ground calibration error of GRACE-FO satellites and 1 g/0 g effects after the satellite was in orbit do not exceed 400 μm.

**Table 8.** Results of GRACE-FO satellites' first CM-offset estimation.

| CM-Offset | dx (μm) | dy (μm) | dz (μm) |
|-----------|---------|---------|---------|
| GRACE-C | 37.6 ± 1.1 | −333.1 ± 0.9 | 10.0 ± 1.3 |
| GRACE-D | −56.7 ± 2.5 | −272.8 ± 2.2 | 7.7 ± 2.5 |

However, as shown in Table 4, the "+103 μm" of the first CM-trim of GRACE-C was performed on 18 July 2018, indicating that the first CM-offset of GRACE-C that has been calculated in this paper is somewhat overestimated. During the estimation of the CM-offset on 6 June 2018, the linear acceleration in the Z direction, which was measured by ACC during two roll maneuvers, was abnormal, therefore, the CM-offset in the Y direction could not be accurately estimated, and only the pitch and yaw maneuvers at low-latitudes were used to estimate the CM-offset in the Y direction.

However, the CM-trim of GRACE-D has not been carried out since its launch, which may be due to the abnormal operation of the GRACE-D ACC. We tried to calculate the CM-offset of GRACE-D on 30 October 2018, and found that the abnormal linear acceleration led to an abnormal estimated CM-offset, therefore, there was no subsequent estimation of the CM-offset of GRACE-D.

The same estimation algorithm was used to calculate the CM-offset of GRACE-C since its launch and the CM-offset estimation curve, as shown in Figure 12, and the CM-offset estimation results were obtained, as shown in Table 9. The two red lines in Figure 12 indicate the specified range of the CM-offset, indicating that the ACC of the GRACE-C satellite can work normally.

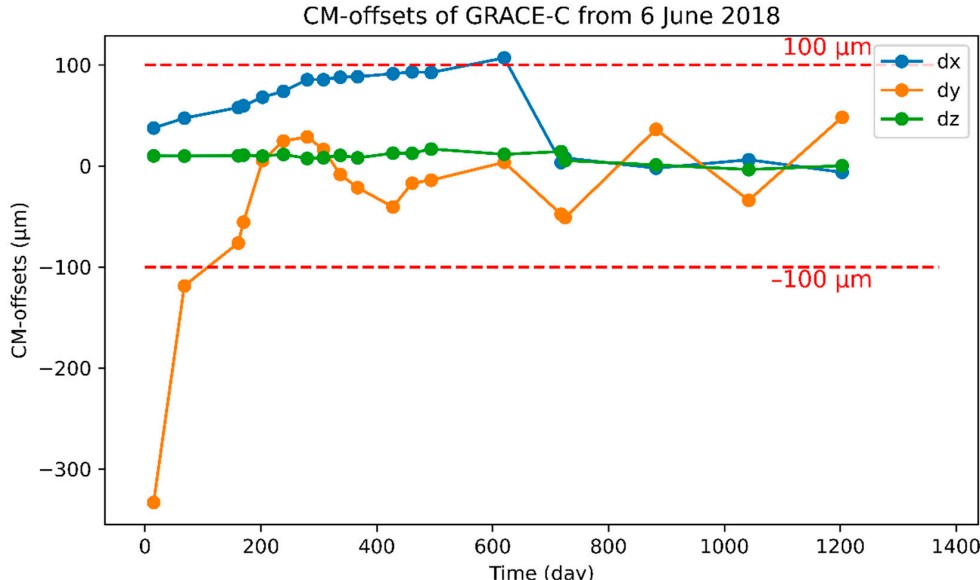

**Figure 12.** Estimation of CM-offset of GRACE-C.

**Table 9.** Estimation of CM-offset of GRACE-C.

| Date | Days | dx (μm) | dy (μm) | dz (μm) |
|---|---|---|---|---|
| 6 June 2018 | 15 | 37.6 ± 1.1 | −333.1 ± 0.9 | 10.0 ± 1.3 |
| 29 July 2018 | 68 | 47.3 ± 0.7 | −118.7 ± 0.2 | 9.9 ± 0.1 |
| 30 October 2018 | 161 | 57.9 ± 0.8 | −76.2 ± 0.2 | 10.2 ± 0.1 |
| 8 November 2018 | 170 | 59.6 ± 0.3 | −55.6 ± 0.2 | 10.8 ± 0.1 |
| 11 December 2018 | 203 | 68.1 ± 0.5 | 5.3 ± 0.2 | 9.7 ± 0.1 |
| 16 January 2019 | 239 | 74.0 ± 0.3 | 24.8 ± 0.1 | 11.5 ± 0.1 |
| 26 February 2019 | 280 | 85.4 ± 1.9 | 28.8 ± 0.2 | 7.5 ± 0.1 |
| 26 March 2019 | 308 | 85.5 ± 1.3 | 16.3 ± 0.2 | 8.2 ± 0.1 |
| 24 April 2019 | 337 | 87.9 ± 0.7 | −8.4 ± 0.5 | 10.5 ± 0.2 |
| 24 May 2019 | 367 | 88.4 ± 0.5 | −21.5 ± 0.2 | 8.0 ± 0.1 |
| 24 July 2019 | 428 | 91.3 ± 0.5 | −40.6 ± 0.2 | 12.7 ± 0.1 |
| 26 August 2019 | 461 | 93.2 ± 0.8 | −17.0 ± 0.4 | 12.6 ± 0.1 |
| 28 September 2019 | 494 | 92.5 ± 1.6 | −14.1 ± 0.2 | 16.8 ± 0.1 |
| 1 February 2020 | 620 | 107.0 ± 0.4 | 3.6 ± 0.2 | 11.5 ± 0.1 |
| 9 May 2020 | 718 | 3.4 ± 0.6 | −47.7 ± 0.3 | 14.3 ± 0.1 |
| 16 May 2020 | 725 | 7.9 ± 0.3 | −51.0 ± 0.2 | 5.4 ± 0.1 |
| 20 October 2020 | 882 | −2.1 ± 0.8 | 36.3 ± 0.1 | 1.0 ± 0.1 |
| 29 March 2021 | 1042 | 6.2 ± 0.6 | −34.0 ± 0.1 | −3.5 ± 0.1 |
| 6 September 2021 | 1203 | −6.3 ± 0.9 | 48.1 ± 0.2 | 0.2 ± 0.1 |
| 20 February 2022 | 1370 | −22.5 ± 2.5 | −33.4 ± 0.3 | 5.9 ± 0.1 |

The values that are marked in red in Table 9 indicate that the CM-trim events occur between two CM-offset estimates. According to the CM-offset estimation results of 29 July 2018, after the first CM-trim on 18 July 2018, there is still a CM-offset of −118 μm in the Y direction, indicating that the first CM-offset result of GRACE-C, as shown in Table 8, has a bias of about 100 μm and, therefore, still needs to be investigated.

The subsequent estimation results show that the CM-offset in the Y direction gradually decreases and is close to 0 by 11 December 2018, but no CM-trim event occurs during this period. This phenomenon may be related to the continuous stress release after the satellite's initial orbit entry, which is worth further study. After 1 February 2020, the CM-offset in the Y direction fluctuates greatly, while the CM-offset in the X and Z directions fluctuates little. This is because, after 1 February 2020, the estimation of the center of mass is conducted every six months, resulting in a large fluctuation in the CM-offset in the Y direction.

## 5. Discussion

The maneuvering strategy that is shown above can be used to estimate the CM-offset of GRACE-type satellites. According to the calibrated results of the seven maneuvers, the accuracy of the calibration results that are obtained at low latitudes is obviously lower, therefore, the maneuvers at low latitudes should perhaps be given up in order to avoid them affecting the normal inter-satellite pointing, which is necessary when the satellite is working properly.

Since it is difficult to obtain the original (Level-0) ACC observation data, it was difficult to know whether the ACC measurements were calibrated. Therefore, four kinds of angular acceleration were used simultaneously to estimate the CM-offset, which can improve the integrity of the estimation results.

In order to improve the estimation accuracy of the CM-offset, the magnetometer calibration parameters and the satellite moment of the inertia correction parameters can be estimated simultaneously during attitude dynamics fitting. However, based on the judgment of the existing fitting effect and reductions in the complexity of the algorithm, this method was not adopted in this paper.

The shape of the GRACE-FO is 3.2 m in length (spacecraft X axis), 1.9 m in width (spacecraft Y axis), and 0.8 m in height (spacecraft Z axis) and has the same outer dimensions as GRACE [9]. This is part of the explanation for Table 9 shows the Y direction CM-offset fluctuates more than the X direction. However, the fluctuations in the Z direction CM-offset are small, and the explanation for this phenomenon requires a detailed understanding of the internal structure of the satellite, which is beyond the scope of this paper.

## 6. Conclusions

In this paper, the CoM calibration algorithm of GRACE-type gravity satellites is summarized, several calculation methods for angular acceleration are derived, and the CM-offset results that were based on four kinds of angular acceleration were verified using GRACE-FO satellites data measured on 1 February 2020. At the same time, the algorithm was applied to the maneuver data of GRACE-FO's first CoM calibration on 6 June 2018, which showed that the estimation result of GRACE-FO's first CM-offset was about 300 μm, which was close to GRACE's first CM-offset result in April 2002. Subsequently, the CM-offset of GRACE-C since its launch is estimated. The conclusions are as follows:

(1) Magnetometer measurements are in good agreement with the IGRF13 geomagnetic model, which can be used to replace the magnetometer measurements in CoM calibration calculations;

(2) (The attitude dynamics fitting method based on the star camera quaternions that was adopted in this paper has high precision and can be used to calculate the MTQ angular velocity and the angular acceleration;

(3) The ACC angular acceleration can be correctly calibrated by MTQ angular acceleration. Although the calibration results that were obtained by the ACC angular acceleration (calibrated) on 1 February 2020 are in good agreement with those that were obtained by JPL, all four of the results that were obtained by four kinds of angular acceleration have the same level of accuracy;

(4) By comparing the four kinds of angular acceleration, we can find abnormal situations in the angular acceleration, and the optimal estimation accuracy can be obtained by estimating the CM-offset based on the rest of the angular acceleration.

In this paper, the use of the MTQ angular velocity, the ACC angular acceleration, the ACC angular acceleration (calibrated), or the MTQ angular acceleration is recommended to estimate the satellite CM-offset for subsequent GRACE-type gravity satellites. The estimations of the CM-offset of GRACE-type satellites using the total least squares method in order to obtain better estimation accuracy will be discussed in our next study.

**Author Contributions:** Conceptualization, Z.H. and L.C.; Data curation, Z.H.; Formal analysis, Z.H. and L.C.; Funding acquisition, S.L.; Investigation, S.L. and D.F.; Methodology, Z.H. and L.C.; Project administration, S.L. and L.H.; Resources, S.L.; Software, Z.H. and L.C.; Supervision, S.L. and L.C.; Validation, Z.H.; Visualization, D.F.; Writing—original draft, Z.H.; Writing—review and editing, S.L., D.F., and L.H. All authors have read and agreed to the published version of the manuscript.

**Funding:** This study was funded by the National Natural Science Foundation of China (No. 42174007).

**Data Availability Statement:** The data used to support the findings of this study are available from the corresponding author upon request.

**Acknowledgments:** We are very grateful to the GFZ for providing GRACE-FO Level-1A and Level-1B data.

**Conflicts of Interest:** The authors declare no conflict of interest.

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
