# Peer review of "Estimation of the Center of Mass of GRACE-Type Gravity Satellites"

_remotesensing, doi:10.3390/rs14164030_

Round 1

Reviewer 1 Report

The CoM calibration maneuvers scheme and CoM estimation methods of GRACE-type satellites are summarized systematically. The method of calibration ACC angular acceleration by “MTQ angular acceleration” is proposed, and four CM-offset estimation methods are used to estimate the satellite's center of mass simultaneously in the manuscript. The CM-offset results of GRACE-C is estimated in the scientific sense. The English writing should be further polished.

[1]       Page 1, line 24: What about ‘four CM-offset results’ in detail?

[2]       Page 1, line 31: What about Chinese-GRACE satellite?

[3]       Page 2, figure 1: The figure should be removed.

[4]       Page 2, figure 2: What WWW the figure from?

[5]       Page 3, line 72: How to define SF in detail? How about its origin point?

[6]       Page 3, line 74: LOS (Line-of-Sight) -> line of sight (LOS).

[7]       Page 3, line 88: How to express CoM and center of ACC Proof Mass in SF?

[8]       Page 4, line 124: Wang F -> Wang et al. (2010) and Wang (2003).

[9]       Page 4, line 130: Li H -> Li (2009).

[10]   Page 4, line 134: Wang Benli -> Wang et al. (2010). Xin Ning -> Xin et al. (2013).

[11]   Page 5, figure 5: Where is the figure from?

[12]   Page 5, line 172: The origin is defined as the actual CoM. But CoM is changing. So SF may not be suitable for expressing CoM-offset.

[13]   Page 7, line 222: The maneuver may change CoM. What is the residual force? Why is the residual force of long period characteristics?

[14]   Page 7, line 226: What is linear drift signal??

[15]   Page 7, lien 231: What is a low-pass filter?

[16]   Page 7, line 232: Noises may include white noise and colorful noise. How to process colorful noise?

[17]   Page 10, line 326: How to define optimally the weight matrix?

[18]   Page 11, section 3: How to validate the manuscript’s results?

[19]   Page 15, figure 11: What method used to fit ACC?

[20]   Page 15, table 6: How to validate the results in the table? Please explain the basic law and mechanism.

[21]   Page 18, figure 13: What filtering method used? What fitting method used?

[22]   Page 18, table 6: How about the results of JPL?

[23]   Section references: The writing format should be normalized.

Reviewer 2 Report

The paper is very well written and clear. However, the paper seems contain many well known materials, making it difficult to see the original contribution of the article. The only original idea of the article is the use of MTQ for the center of mass calibration of a satellite gravity mission. 

Some questions are in need of answers from the authors. 

1. No mention of whether manuevur is needed in the calibration. Is the method applicable in the commissioning phase or scientific phase of the mission? 

2. Though the data of star sensor spans over the entire frequency band of gravity field measurement, the most sensitive part is in the sub mHz band which lies outside the measurement band. While the angular acceleration measurement of an accelerometer is most sensitive in around 0.01Hz or higher. Does it make sense to calibrate angular signal of an accelerometer using star sensor? 

3. There is a need to calibrate an accelerometer signal at the level of 10^-9ms^{-2}/\sqrt Hz. Is the MTQ with magnetic field generated by mA current sensitive enough for this job? The glitches in the ACC 1A data seems to be against this idea. 

4. The problem of MTQ is the residual magnetic field of the spacecraft itself and the spacecraft charging that generates the Lorentz force acting on the 

charged payload. This part is not calibrated or measured in the commissioning phase of the mission. How do you account for the bias of the MTQ measurement from this noise source. 

5. For attitude estimate, there is a standard algorithm to combine the data from the star sensor and the accelerometer, with weight given to the respective data. This works quite well in the attitude estimate. Perhaps similiar algorithm should be considered to combine different data stream available? This is quite a standard practice in navigation. See for instance Journal of Geodesy volume 93, pages 1881–1896 (2019).

6. The standard COM calibration using cold gas is quite well established and free of many noise sources mentioned above. I do not see any reason or advantage of changing it. The amount of cold gas is more than sufficient for this calibration purpose. What is the motivation or need behind this suggested new algorithm.
